# High-resolution hydroacoustic datasets for over 7000 km² of Southern Baltic

Łukasz Janowski[1], Anna Barańska[1], Aleksandra Bojke[1], Roksana Bona[2], Anna Borecka[1], Agnieszka Cichowska[1], Grażyna Dembska[1], Patryk Dombrowski[2], Diana Dziaduch[1], Agnieszka Flasińska[1], Sara Foit[2], Aleksandra Gadzińska[2], Łukasz Gajewski[2], Juliusz Gajewski[1], Katarzyna Galer-Tatarowicz[1], Karol Ginał[2], Urszula Grzywińska[2], Izabela Górecka[2], Olha Hruzdieva[2], Edyta Jurkiewicz-Gruszecka[3], Natalia Kaczmarek[1], Marcin Kalarus[1], Magdalena Kamińska[3], Maciej Kałas[1], Sandra Korczak[2], Mateusz Kołakowski[2], Maria Kubacka[1], Tomasz Kusio[1], Emilia Leszczyńska[2], Aliaksandr Lisimenka[1], Małgorzata Marciniewicz-Mykieta[3], Monika Michałek[1], Ewelina Misiewicz[1], Jarosław Nowak[2], Radosław Opioła[1], Piotr Pieckiel[1], Karolina Rogowska[2], Marcin Sontowski[2], Marta Szafrańska[2], Kazimierz Szefler[1], Anna Tarała[1], Radosław Wróblewski[1,2], Paweł Wysocki[2], Krzysztof Załęski[1]

[1]Maritime Institute, Gdynia Maritime University, Roberta de Plelo 20, 80-548 Gdańsk, Poland
[2]MEWO S.A., Starogardzka 17A, 83-010 Straszyn, Poland
[3]Environmental Monitoring Department, Chief Inspectorate for Environmental Protection, Bitwy Warszawskiej 1920 r. 3, 02-362 Warszawa

*Correspondence to*: Łukasz Janowski (ljanowski@im.umg.edu.pl)

**Abstract.** This study presents high-resolution hydroacoustic datasets covering over 7,000 km² of Polish Marine Areas in the Southern Baltic, acquired between April 2022 and December 2023 as part of a national initiative to map benthic habitats using advanced sonar technologies. Utilizing a fleet of seven vessels and the expertise of approximately 250 personnel, the project collected bathymetric and side-scan sonar data along more than 95,000 km of survey lines, adhering to International Hydrographic Organization S-44 Order 1a standards. The resulting datasets include detailed bathymetric grids at 50 x 50 cm resolution and side-scan sonar backscatter mosaics at 20 x 20 cm resolution, with robust quality control ensuring at least 95% data completeness per grid cell. These data provide unprecedented insight into the underwater topography and sediment characteristics of the region, supporting applications in scientific research, environmental management, offshore wind farm planning, and underwater archaeology. The datasets, available at DOI: https://doi.org/10.26408/southern-baltic-hydroacoustic-datasets, lay a solid foundation for future studies and the development of science-informed policies to promote sustainable and resilient marine ecosystems in the Baltic Sea.

## 1 Introduction

Mapping our seas and oceans is essential for understanding and protecting marine environments, supporting sustainable resource management, and enhancing navigation safety. The United Nations Decade of Ocean Science for Sustainable

Development (2021-2030), known as the Ocean Decade, aims to generate critical ocean knowledge to manage the ocean sustainably (Wu et al., 2020). High-resolution mapping provides valuable data for studying ocean currents, marine biodiversity, and underwater hazards, which is crucial for addressing climate change and its impacts (Wölfl et al., 2019). By improving our understanding of the ocean, we can develop science-informed policies to maintain a resilient and productive marine ecosystem for future generations (Wölfl et al., 2019).

Mapping the seafloor in high resolution using hydroacoustic devices is essential for several reasons, particularly in the context of offshore wind farms, sustainable development, benthic habitat identification, and underwater archaeology. For offshore wind farms, detailed seafloor maps are essential for safe and efficient placement of turbines, minimizing environmental impact and optimizing energy production (Varnell et al., 2023). High-resolution mapping supports sustainable development by providing critical data for managing marine resources and protecting ecosystems (Buhl-Mortensen et al., 2015). Benthic habitat mapping helps to identify and monitor diverse marine habitats, aiding in conservation efforts (Lacharité and Brown, 2019). Additionally, hydroacoustic mapping is invaluable for underwater archaeology, revealing submerged cultural heritage sites and shipwrecks, thus preserving historical and archaeological treasures (Papadopoulos, 2021; Janowski et al., 2024). Overall, these maps enhance our understanding of the ocean, supporting various scientific, environmental, and economic activities.

The Seabed 2030 project, a collaboration between The Nippon Foundation and GEBCO, aims to have the entire ocean floor mapped by 2030, providing high-resolution bathymetric data to support sustainable ocean management (Mayer et al., 2018). This initiative is particularly significant for the Baltic Sea, a region with complex underwater topography and diverse ecosystems. Mapping the seafloor in high-resolution using hydroacoustic devices is also crucial for achieving the goals of the Marine Strategy Framework Directive (MSFD), which aims to protect marine ecosystems and achieve good environmental status (GES) of EU marine waters (Maes et al., 2018). High-resolution seafloor maps provide essential data for monitoring seabed integrity, biodiversity, and the impacts of human activities, such as fishing, dredging and dumping (Galparsoro et al., 2015). This information supports the sustainable management of marine resources, helps identify and protect sensitive habitats, and ensures compliance with environmental regulations (Smith et al., 2016). Additionally, detailed seafloor mapping aids in the assessment and mitigation of underwater hazards, contributing to safer maritime navigation and operations.

A regional response to MSFD requirements was the Polish national project including the performance of high-resolution hydroacoustic measurements in the southern Baltic Sea. The project titled: "Mapping of Benthic Habitats of Polish Marine Areas using the sonar mosaicking method in 2021–2023" was established by the Polish Chief Inspectorate for Environmental Protection within the framework of the State Environmental Monitoring and financed from the Polish national budget. The hydroacoustic datasets were carried out by a consortium of the Maritime Institute of Gdynia Maritime University and MEWO S.A company.

Over 1.5 years, a fleet of seven vessels and approximately 180 personnel were dedicated to acquiring high-resolution hydroacoustic datasets in the Polish part of the Southern Baltic. An additional 30 people focused on data processing, while

40 were responsible for project coordination and logistics. This extensive effort marked the first comprehensive exploration of the seafloor in this area, covering over 7300 km². It involved over 95,000 km of survey lines and spending around 500 days (12,000 hours) at sea. The study provided invaluable initial insights into the region's underwater topography and

habitats, laying the groundwork for future research and sustainable management of marine resources.

## 2 Study area

According to the project requirements, the hydroacoustic survey area encompassed the region outlined by the Spatial Plan of the Polish Maritime Areas at a scale of 1:200,000. However, certain areas were excluded from the survey: the immediate vicinity of the shore up to the 5-meter isobath, depths greater than 60 meters, military-designated zones, offshore wind

energy-designated areas where previous hydroacoustic surveys had been conducted, and areas designated as having minimal or no benthic habitat value according to environmental designations. The spatial datasets were required to have at least one actual bathymetric measurement present in 95% of the pixels of the resulting raster grids, with pixel sizes of 50 x 50 cm for multibeam echosounder (MBES) or 20 x 20 cm for side-scan sonar (SSS) measurements.

The project assumed the acquisition of high-resolution hydroacoustic datasets for the survey area of at least 7318.47 km².

Within this area, bathymetric and SSS measurements were conducted. Additionally, data were sourced from our previous high-resolution measurements conducted in Polish marine areas, specifically, on the Słupsk Bank and Koszalin Bay. Where possible, MBES bathymetry from these measurements was reprocessed to ensure consistency with the newly acquired hydroacoustic datasets. However, some legacy datasets differ in resolution and do not include MBES backscatter, introducing heterogeneity. These datasets were included to maintain spatial completeness but should be used with caution for

comparative or quantitative analyses. These datasets are listed and marked in Table 1 and Figure 1 below.

**Table 1.** MBES and SSS datasets with non-standard resolutions or incomplete coverage, including data from previous surveys (Słupsk Bank and Koszalin Bay) integrated into the overall dataset.

| Folder/file name | Resolution [m] |
| --- | --- |
| MBES/N3357B_N3357C_N3357D_N3358A | 1 |
| MBES/N3334C_N3334D_N3335C_N3345B_N3346A_N3346B_N3347A_18 | 2 |
| SSS/N33333D_N3334C_N3334D/[all files] | 0.1 |
| SSS/N3335C_N3335D_N3347A_N3347B/[all files] | 0.1667 |
| SSS/N3357B_N3357D/398_IIA_plus | 1 |
| SSS/N3334C_N3334D_N3335C_N3345B_N3346A_N3346B_N3347A/LS_Glazowisko_SSS | 1 |

## 2.1 Separation for study sheets

The surveys were systematically executed by dividing the whole geophysical survey area and the habitat delineation area into survey sheets at a scale of 1:50.000. These were designated as survey sheets for the project. A total of 101 survey sheets were created and uniquely numbered, facilitating the identification of both the geophysical survey and study areas (see Figure 1). The division and nomenclature of these sheets followed the Ordinance of the Polish Council of Ministers of October 15, 2012, on the state system of spatial references (Journal of Laws 2012, item 1247). This approach also allowed for the successive acquisition of geophysical data in finite areas presenting similar environmental conditions.

For example, survey sheet N-13-33-C (denoted as N1333C) covers the area bounded by coordinates 17°30' E, 55°40' N and 17°45' E, 55°50' N. Within each survey sheet, latitudinal (W-E direction) survey profiles were set 0.005 nautical miles (approximately 9.26 meters) apart. This resulted in 2,000 profiles. The Party Chief conducting the surveys selected the subsequent profiles to be surveyed based on the range of the survey equipment used (MBES and SSS) to ensure full coverage of the hydroacoustic surveys.

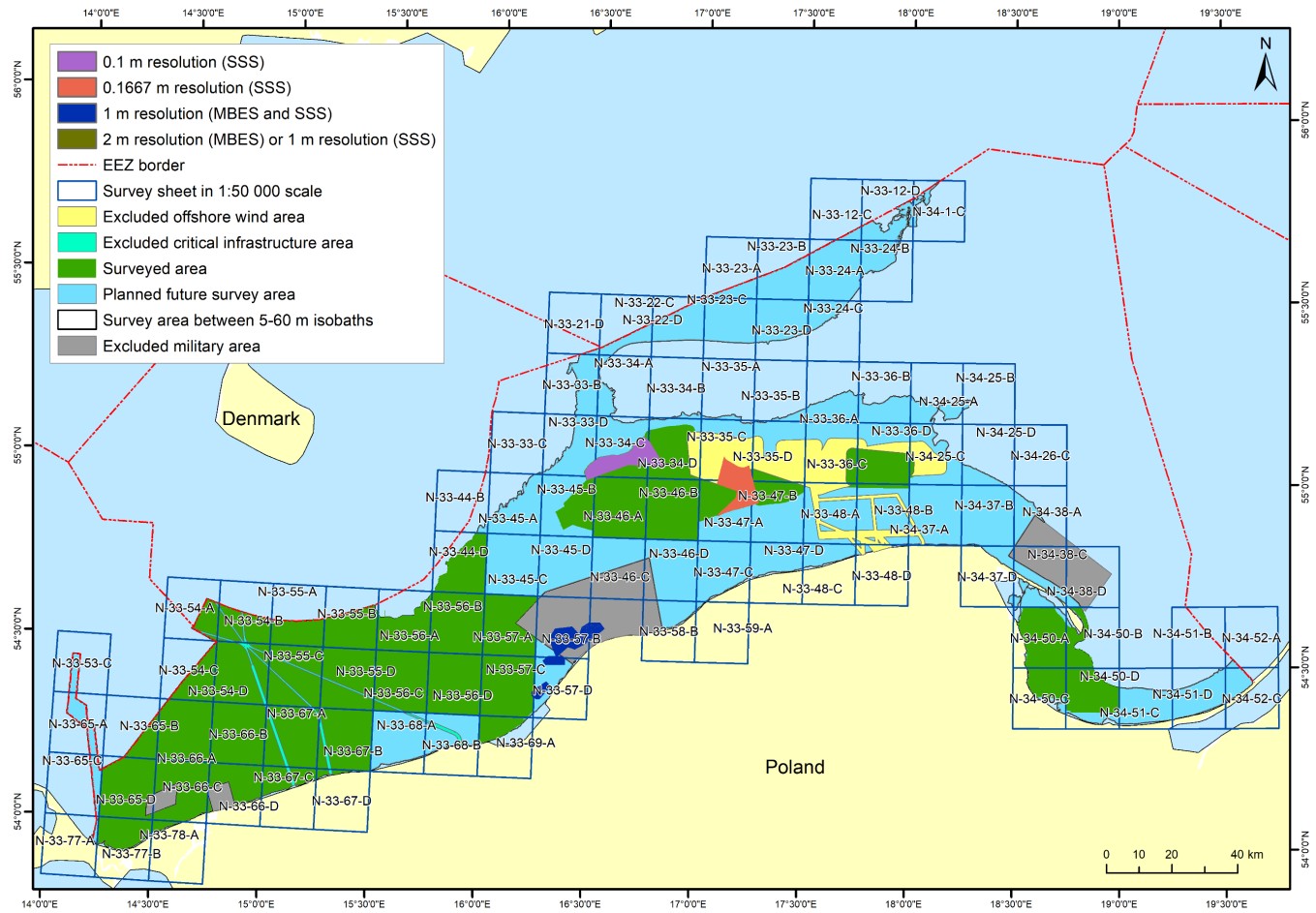

**Figure 1. The study area divided into 1:50,000 scale survey sheets, detailing specific zones such as hydroacoustic study areas, offshore wind energy areas, critical infrastructure, military infrastructure, and areas designated for future potential surveys.**

## 3 Materials and methods

### 3.1 Hydroacoustic data acquisition

Bathymetric and SSS surveys were performed in the period from 28 April 2022 to 5 December 2023; from aboard the

following seven vessels: FRITZ REUTER, HELOT, HYDROCAT TAPPER, IMOR, IMOROS 2, MIMER, SITULA. Table 2 provides a summary of all vessels conducting data acquisition with their specifications and measuring equipment (MMSI - Maritime Mobile Service Identity). All measurements were acquired within the PL-EVRF2007-NH height reference level and ETRS89/Poland CS92 reference system based on the WGS 1984 ellipsoid. Positioning of vessels and their measurement systems was conducted using the following approaches: RTK DGPS positioning with the ASG EUPOS correction system,

inertial positioning with motion, heading, and acceleration sensors, and USBL (Ultra-Short Baseline) underwater

positioning. All hydroacoustic data acquisition surveys were performed in accordance with the IHO (International Hydrographic Organization) S-44 1a order (International Hydrographic Organization, 2020, 2022).

**Table 2.** Summary of vessels with their properties and the measuring equipment used. All dimension measures are expressed in meters

| Vessel | Length | Width | Draught | MMSI | MBES | SVP | SSS |
|---|---|---|---|---|---|---|---|
| FRITZ REUTER | 24 | 7 | 2.7 | 211223180 | Reson T50 | SVP 70 | Edge Tech 4205 |
| HELOT | 30 | 8 | 3.5 | 261005510 | R2Sonic 2026 | MiniSVS | Edge Tech 4205 |
| HYDROCAT TAPPER | 22 | 6 | 1.7 | 261000338 | Reson T50 | SVP 70 | Edge Tech 4205 |
| IMOR | 33 | 11 | 2.7 | 261379000 | Reson T50 | SVP 70 | Edge Tech 4205 |
| IMOROS 2 | 11 | 3.5 | 0.65 | 261012680 | Reson T50 | MiniSVS | - |
| MIMER | 27 | 7 | 3.7 | 311000862 | Reson 7125 | SVP 70 | Edge Tech 4205 |
| SITULA | 38 | 9.5 | 3.4 | 356151000 | Reson T50 | SVP 70 | Edge Tech 4205 |


To achieve accurate survey outcomes, measurements incorporated controls for environmental conditions (sea state, vessel stability, equipment setup). The measurements were conducted only in a calm sea state of no more than 3, according to the Beaufort scale (slight – waves with heights of 0.0 to 1.25 meters). Sound velocity profiles were acquired at minimum six-hour intervals and more frequently when bathymetric data showed thermocline or refraction effects.

The project's quality requirements stipulated that at least one actual bathymetric measurement should be in 95% of the pixels of the resulting raster grids (with pixel sizes of 50 x 50 cm or 20 x 20 cm for SSS measurements). Maximum vessel speeds were set at 7 knots for MBES-only surveys and 4.1 knots for combined MBES/SSS surveys (at 65 m sonar range), with operators adjusting speeds to maintain ≥20 measurement points per meter across-track and ≥5% swath overlap.

### 3.1.1 Multibeam data acquisition

Bathymetric surveys with a MBES were carried out in a way that ensured full coverage of the seabed with a 10% overlap. Measurements were conducted along the survey profiles planned in individual survey sheets. For the entire survey area, the initial survey profiles were designed every approx. 9.26 meters from one another. During the measurements, on an ongoing basis, the Party Chief of each unit, based on the current depth of the surveyed body of water, decided what line should be chosen to meet full data coverage. In places where the coverage of the seabed with bathymetric data was not sufficient,
supplementary profiles were added and surveyed. In addition, during bathymetric measurements, MBES backscatter data were also acquired.

The primary measurement systems used to create a bathymetric image of the seabed in the surveyed area were Teledyne Reson 7125, T-50, and R2Sonic 2026 MBESs. Teledyne Reson 7125 was primarily used for filling gaps in data coverage. The software for data acquisition and recording in real time was QPS QINSy 9.3. DGPS RTK systems were used for the

positioning of the survey units. Elevation correction to the PL-EVRF2007-NH vertical elevation system was based on ASG-EUPOS virtual reference stations. Settings of all MBES devices in all vessels were kept in the same values throughout the whole survey. Table 3 summarizes the precise values of the used parameters.

To ensure compliance with IHO S-44 Order 1a accuracy standards, the survey vessels underwent regular validation and calibration procedures. Before each measurement mission, all vessels performed system testing and verification using reference points with established coordinates and depths located in Gdańsk Bay near Gdynia port. Specifically, two reference stones with precisely known positions and elevations were used for vertical and horizontal accuracy verification. Additionally, pitch and roll calibration parameters were checked on known reference targets including submerged objects of established position such as the "Desantowiec" landing craft and the Franken shipwreck.

Cross-check validation was performed by running control profiles through multiple survey sections to verify line-to-line consistency and identify potential systematic errors. However, comprehensive statistical analysis of all validation data across the entire 7,300 km² survey area was not conducted as part of this dataset publication. The validation procedures performed indicate that the survey met or exceeded IHO S-44 Order 1a requirements, though formal independent verification against external reference surveys was not undertaken.

Table 3. Technical Specifications for Bathymetric survey

| Parameter | Value |
| --- | --- |
| Acoustic signal frequency | During bathymetric surveys, all MBESs should operate at frequencies ranging from 300 to 400 kHz |
| Scanning frequency (ping rate) | The maximum scanning frequency should be 50 Hz |
| Single beam width | At a frequency of 400 kHz, the following beam widths should be achieved:<br>• Along the track: 1°<br>• Across the track: 0.5° |
| Coverage | Bathymetric survey along survey lines should be conducted ensuring a measurement density of 4 points per m² |
| Distance between lines | Ensures minimum area coverage with measurement points (4 points per m²) |
| Overlap of adjacent survey lines (%) | Minimum 10% |
| Data Density per m² | Minimum 4 |
| Grid size | 0.5 m x 0.5 m |
| Ship speed | Up to 7 knots |

### 3.1.2 SSS data acquisition

The survey lines for SSS in the study area were analogous to the basic bathymetric survey lines. The maximum recording width of the SSS swath was no more than 65 m per side. The level of the sonar guidance above the seabed was adjusted to the SSS's operating range to ensure an adequate degree of seabed coverage.

Sonar surveys were carried out with the EdgeTech 4205 SSS devices. The SSS transducer array was towed in a way that allowed its stable position in the water depths. The position of the sonar float was determined in real time using an acoustic positioning system (ultra-short baseline – USBL). Digital data from the SSS were stored in the EdgeTech Discovery sonar data collection and processing software. In very shallow areas, the system was positioned using a fixed offset, the so-called layback method. During the measurements, the Edge Tech 4205 SSS operated at frequencies of 300 kHz and 600 kHz. The primary frequency used to create a large-area mosaic of the sea seabed was 600 kHz.

### 3.2 Hydroacoustic data processing

The processing of MBES data included a bathymetric map (a DEM of the seabed) and, in the case of MBES, backscatter or SSS mosaic data, a sonar mosaic image. Bathymetric data processing consisted of two primary steps: (1) the removal of acoustic noise and erroneous measurements, and (2) the standardization of all depth values to the PL-EVRF2007-NH vertical reference system referenced to mean sea level.

### 3.2.1 Multibeam data processing

MBES bathymetry data processing and quality control were performed in the following three steps:

1. Utilization of QINSy software to analyze and, if necessary, correct the input data acquired from individual devices. Export of output data in *.fau data format.
2. Analysis and filtering of bathymetric data in BeamWorx AutoClean software. This included: RTK corrections and standard filtration techniques: refraction correction (classic and overlap), surface spline, 95% confidence level, outside beam clipping, and auto height fitting.
3. Export of processed data into surface GeoTiff grid rasters (x, y, z).

MBES backscatter data processing and quality control were performed in the following three steps:

1. Analysis and filtering of MBES backscatter data using custom software developed for this project. Processing included Time-Varying Gain (TVG) compensation (logarithmic attenuation correction based on range, beam number, and intensity), spike cleaning, and outlier removal.
2. Export of processed data in backscatter GeoTiff grid rasters (x, y, z).

Because of the limited time of the project and the large amount of data collected, all MBES backscatter datasets were processed to $BL_0$ data format (raw backscatter datasets, without angular compensation correction, (Schimel et al., 2018)).

### 3.2.2 SSS data processing

SSS datasets recorded using EdgeTech Discovery software were processed in SonarWiz software. For a clear and most accurate representation of the structure of the seabed, the highest possible frequency data was used (600 kHz, according to the technical capabilities of the SSS), contained in the third and fourth channels of sonar data recording, which corresponded to high-frequency data.

SSS data processing and quality control were performed in the following two steps:

1. Analysis and filtering of SSS data in SonarWiz software. This analysis included: the correction of navigation errors, image gain normalization, destripping, geometric correlation removing the dead zone under the transducer, and regulation of time-varying gain (TVG).

2. Export of processed data in the SSS GeoTiff image mosaic rasters (x, y, z)

## 4 Results and discussion

The hydroacoustic data acquisition and processing for the project involved detailed timelines and comprehensive surveys using multiple vessels. This section outlines the results and discussion of hydroacoustic data acquisition and processing, focusing on MBES and SSS datasets.

### 4.1 Hydroacoustic data acquisition

The timelines for hydroacoustic surveys by vessel are shown in Figure 2. The detailed timelines for all vessels are summarized in Table S1 in Supplementary Material.

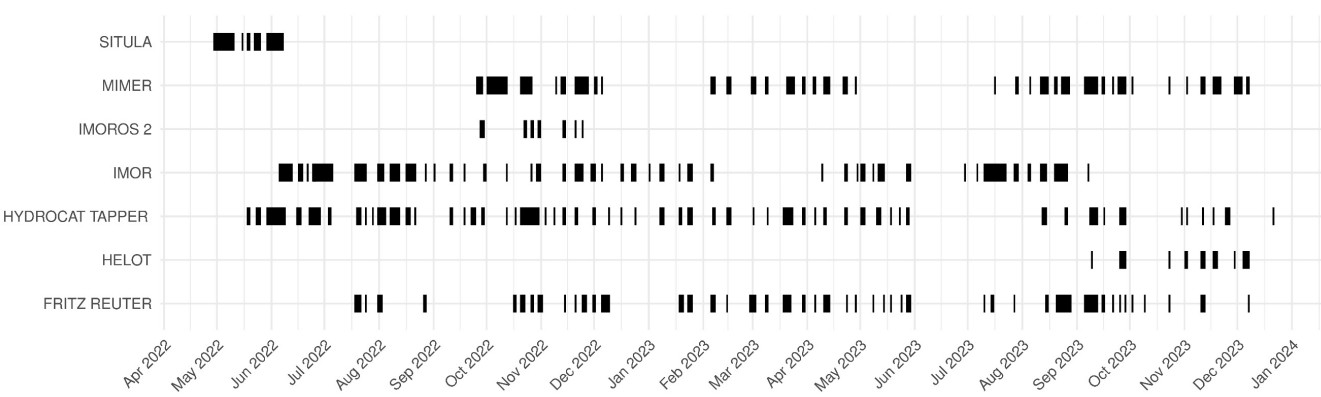

**Figure 2. Timelines for hydroacoustic surveys by individual vessel. The horizontal axis represents the months and years, while the vertical axis indicates the different vessels involved**

## 4.2 Hydroacoustic data processing

The results of MBES and SSS data processing in terms of completeness of data coverage of individual survey sheets are shown in Figure 3. The detailed areas for all survey sheets are provided in Table S2 in Supplementary Material.

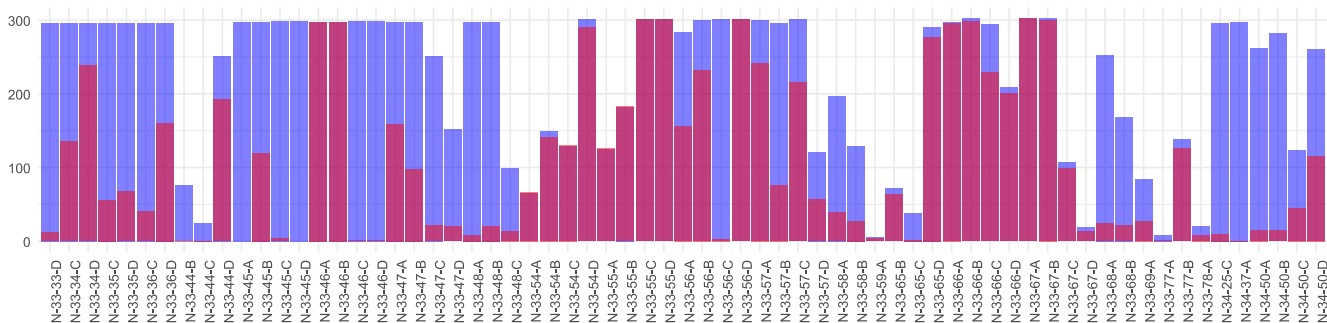

**Figure 3. Summary of surveyed areas with bathymetric and sonar results by survey sheet. The outer columns (purple) indicate the maximum area covered by the hydroacoustic surveys, and the inner columns (burgundy) within these outer columns represent the**
**total surveyed area within the maximum area. The horizontal axis represents the survey sheet, while the vertical axis provides the survey area in km²**

### 4.2.1 MBES dataset

The obtained bathymetric and MBES backscatter data, as results of data processing, were checked for quality and subjected to preliminary geomorphological analysis. The nature of the seafloor geomorphology and the type of sediment covering the
seabed surface were analyzed. Figure 4 and Figure 5 provide examples of a bathymetric map and MBES backscatter data of the same section of the seabed, showing different types of seabed surface.

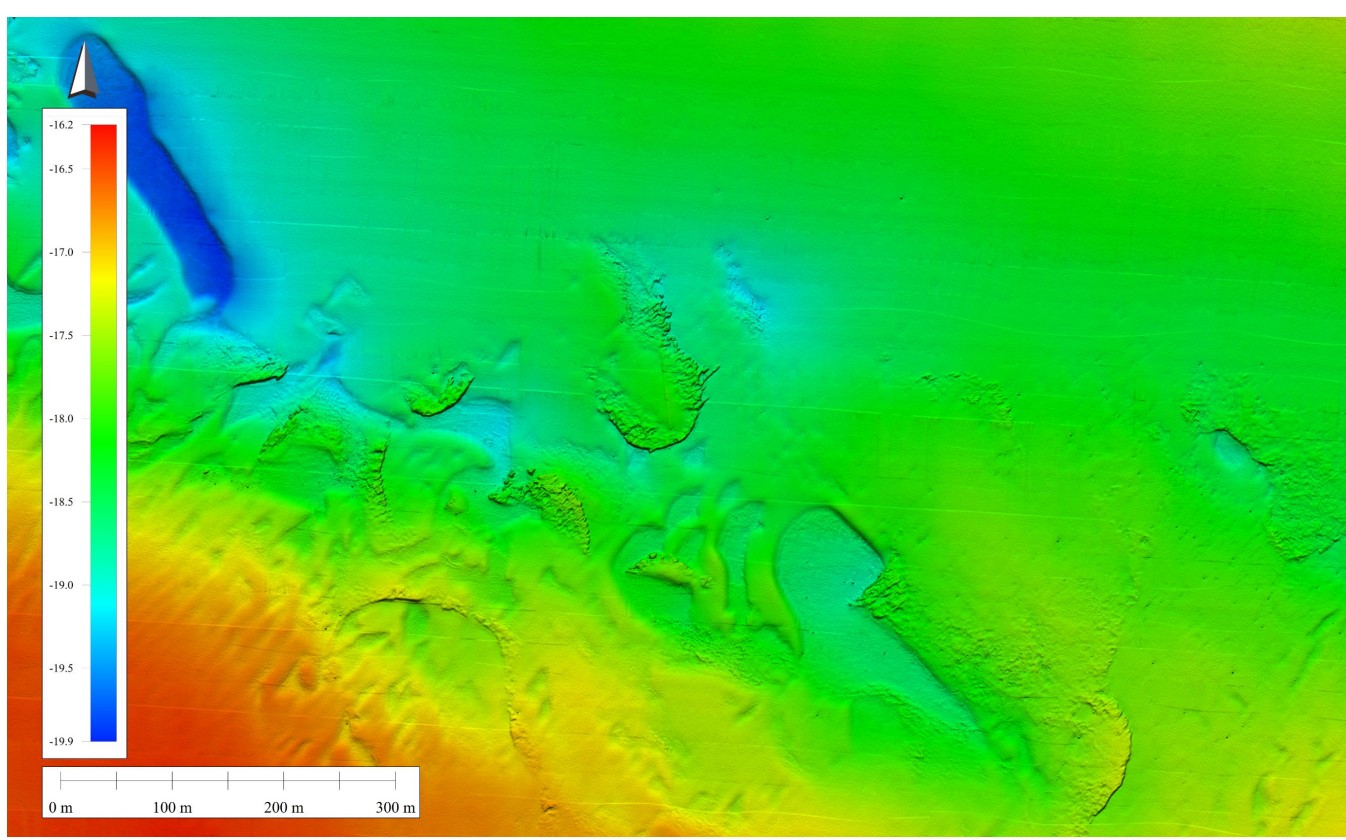

**Figure 4. Fragment of the bathymetric map, survey sheet N-33-66-A; in the central part of the map there are visible outcrops of cohesive sediments with single boulders and sandy seabed in the surroundings**

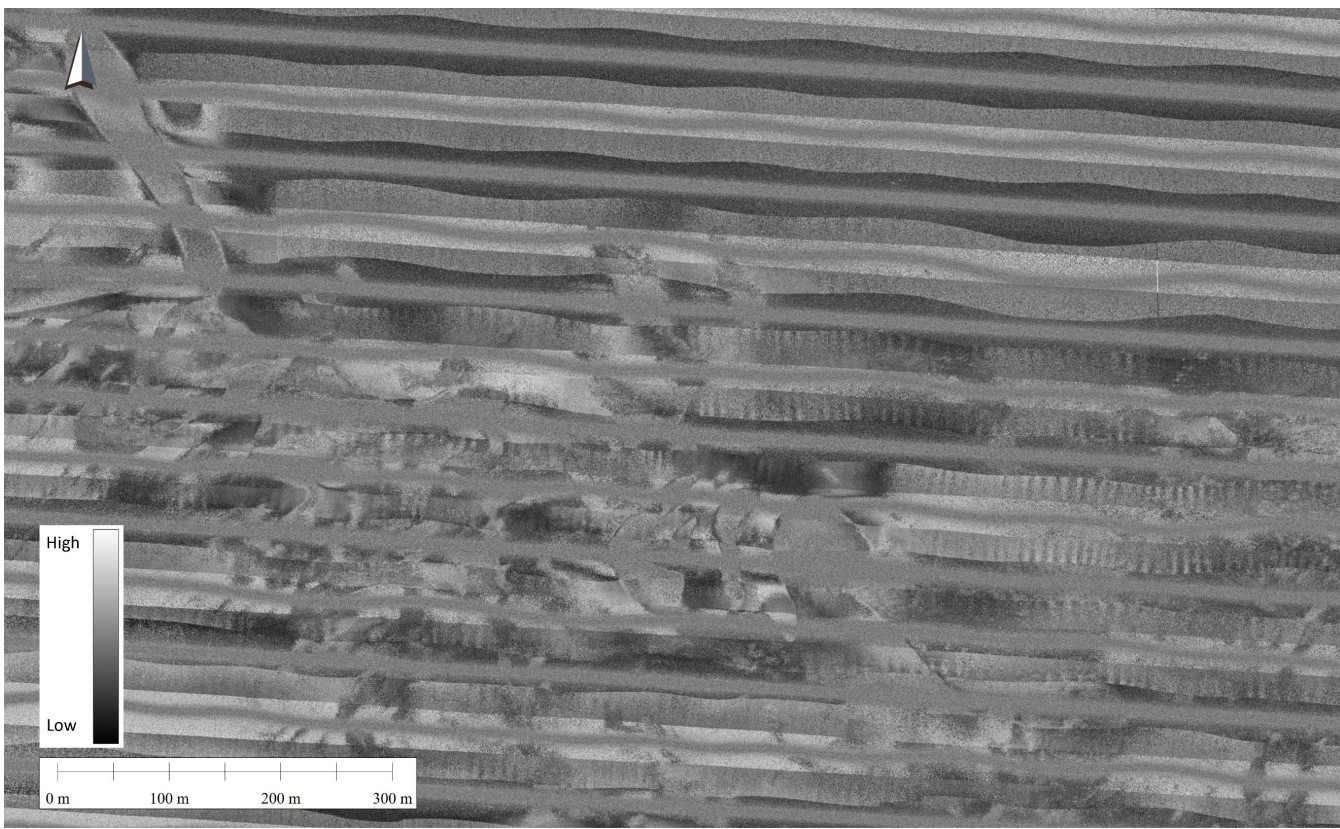


**Figure 5. Fragment of the map with MBES backscatter data, sheet N-33-66-A; in the central part of the map, there are outcrops of cohesive sediments with single boulders and sandy seabed in the surroundings**

### 4.2.2 SSS dataset

Similarly like in the case of MBES measurements, the obtained SSS data were checked for quality and subjected to
preliminary geomorphological analysis. The type of seafloor sediments covering the seabed surface were analyzed. Figure 6 provides an example of a SSS image mosaic of the same section of the seabed like in the previous section, showing different types of seabed surface.

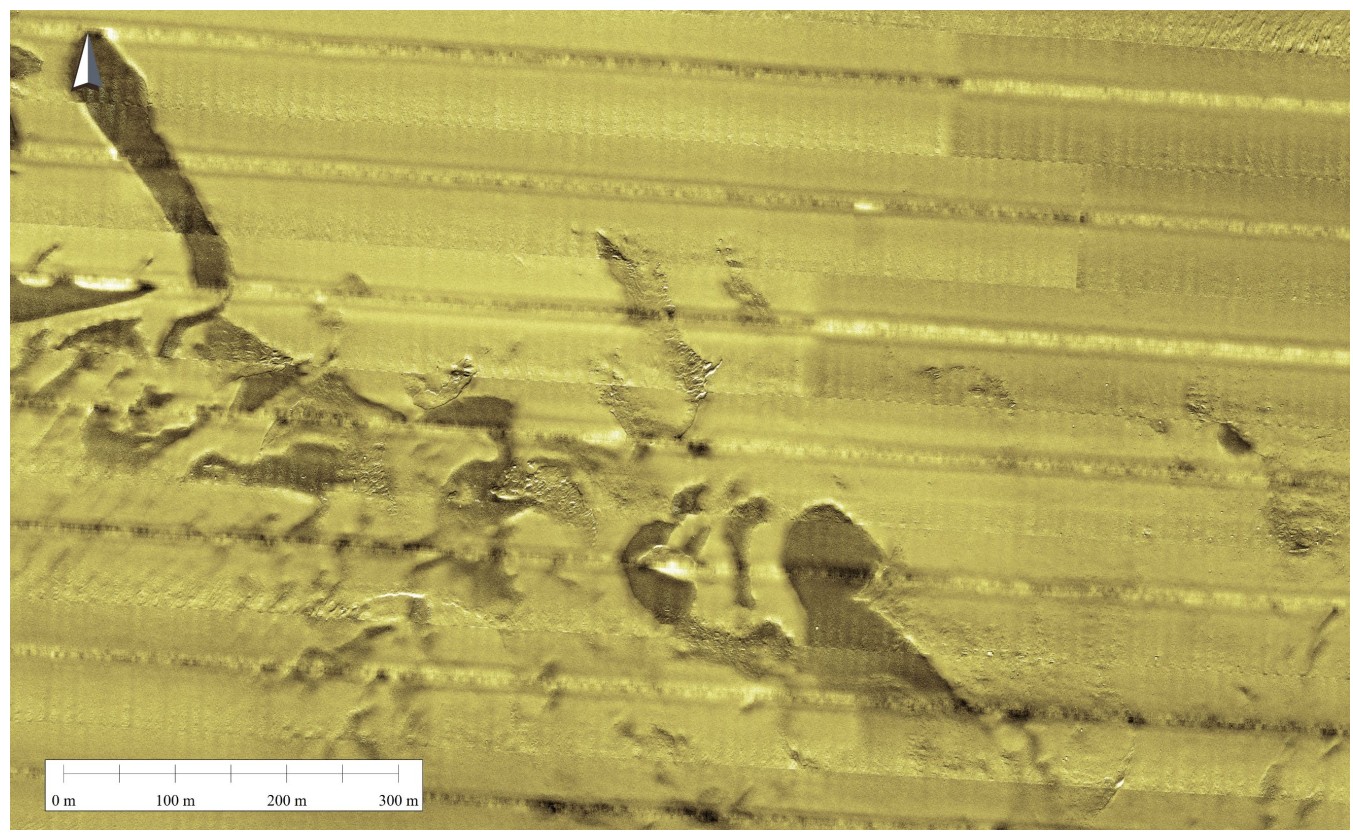

**Figure 6. Fragment of SSS map, sheet N-33-66-A; in the central part of the map outcrops of cohesive sediments with single boulders and sandy seabed in the surroundings**

## 4.3 Technical validation of the datasets

Technical validation of the hydroacoustic datasets was performed through a multi-step process encompassing both acquisition and post-processing stages. All data collection adhered to IHO S-44 Order 1a standards, with survey protocols designed to maximize data quality and minimize uncertainties.

During acquisition, vessel speed was strictly controlled (maximum 7 knots for MBES-only surveys and 4.1 knots for combined MBES and SSS surveys) to maintain data density and avoid spatial gaps. Sound velocity profiles were regularly collected to correct for water column effects, and equipment calibration was routinely verified. Data completeness was ensured by monitoring the percentage of valid measurements per grid cell, with a project requirement of at least 95% coverage.

Post-processing involved several layers of quality control. Bathymetric data were filtered and corrected for refraction, outliers, and noise using QINSy and BeamWorx AutoClean software. Backscatter data underwent spike cleaning and outlier

removal. The processed datasets were exported as GeoTiff grid rasters, facilitating further analysis and integration with other spatial data.

Visual inspections of bathymetric and backscatter maps, as well as SSS mosaics, confirmed the ability to resolve fine-scale seafloor features and sediment types across most of the survey area. The MBES bathymetric data are of high quality and well-suited for quantitative applications including geomorphological analysis, habitat mapping, and detailed seafloor characterization. However, users should note key limitations in the datasets. MBES backscatter data were processed to $BL_0$ format without angular compensation, making them suitable for qualitative sonar verification rather than quantitative

sediment analysis. SSS mosaics show good geometric accuracy but include typical side-scan misalignments of up to ~20 meters; they are best used for qualitative geomorphological and habitat mapping, with MBES bathymetry recommended for precise geometry. Taking these limitations into consideration, the datasets are validated for scientific, environmental, and operational applications including qualitative habitat assessment, regional geomorphological analysis, and marine management planning, while supporting further research and improved understanding of the Polish marine areas.

**4.3.1 Multibeam data quality**

The quality of the MBES dataset was ensured through a combination of rigorous survey planning, adherence to international standards, and comprehensive post-processing. All MBES surveys were conducted in compliance with the IHO S-44 Order 1a standard, which stipulates strict requirements for bathymetric data accuracy and coverage. The survey design provided full seafloor coverage with at least 10% overlap between adjacent survey lines, minimizing data gaps and ensuring

redundancy.

The project utilized high-performance MBES systems, including Teledyne Reson 7125, T-50, and R2Sonic 2026, across seven survey vessels. A summary comparison of the main used MBES systems is provided in Table 4. RTK DGPS positioning and motion reference units were employed to maintain high spatial accuracy. Sound velocity profiles were measured at least every six hours, or more frequently as needed, to correct for water column variability.


Table 4. Comparison of the main MBES parameters used in the project.

| Parameter | Reson T-50 | R2Sonic 2026 | Reson 7125 |
|---|---|---|---|
| Frequency range | 190– 420 kHz | 170–450 kHz (optional 100 kHz, 700 kHz) | 200– 400 kHz |
| Beam width | 0.5° at 400 kHz | 0.45° x 0.45° at 450 kHz | 0.5° x 1° at 400 kHz |
| Maximum depth | 900 m with extended range projector | 800 m+ | 450 m at 200 kHz |
| Ping rate | Up to 50 Hz | Up to 60 Hz | Up to 50 Hz |
| Number of | Up to 1024 | Up to 1024 | Up to 512 |

| | | | |
|---|---|---|---|
| Soundings per Ping | | | |
| Swath coverage | Not specified | 10° to 160° | Up to 165° |
| Integrated INS | Yes | Yes | No |
| Power consumption | Not specified | 100W average | 250W average |
| Operating temperature | -5°C to +45°C (Rack-mounted Sonar Processor) | -10°C to +50°C | -2°C to +35°C |
| Weight | Less than 8 kg in water (sonar head) | Receiver: 12.9 kg, Projector: 13.4 kg | Not specified |

Data quality was quantitatively assessed using key parameters such as THU and TVU. The processed MBES data consistently met or exceeded the IHO S-44 requirements, with THU values typically below 2 meters and TVU values well within the allowable limits for Order 1a surveys. Table 5 presents a summary of MBES system parameters supported calculations of THU and TVU values.

Table 5. Comparison of Uncertainty (THU, TVU) for the primary MBESs by IHO Orders.

| | | Reson T-50 | | R2Sonic 2026 | |
|---|---|---|---|---|---|
| Depth (m) | IHO Order | THU (m) | TVU (m) | THU (m) | TVU (m) |
| 5 | Special Order | 0.05 | 0.02 | 0.045 | 0.02 |
| 10 | Special Order | 0.10 | 0.04 | 0.09 | 0.04 |
| 20 | Order 1a | 0.20 | 0.08 | 0.18 | 0.08 |
| 30 | Order 1a | 0.30 | 0.12 | 0.27 | 0.12 |
| 40 | Order 1b | 0.40 | 0.16 | 0.36 | 0.16 |
| 50 | Order 1b | 0.50 | 0.20 | 0.45 | 0.20 |
| 60 | Order 2 | 0.60 | 0.24 | 0.54 | 0.54 |

The density of measurements exceeded 20 points per meter across the track, ensuring high-resolution bathymetric grids (50×50 cm pixel size). Data completeness was also monitored, with at least one valid depth measurement present in 95% of grid cells, as required by project specifications. Supplementary survey lines were acquired in areas where initial coverage was insufficient, ensuring comprehensive and reliable bathymetric mapping.

Preliminary geomorphological analysis of the MBES data confirmed the ability to distinguish various seafloor features, such as cohesive sediment outcrops, boulders, and sandy substrates. High spatial resolution, precise positioning, and rigorous survey protocols yielded MBES bathymetric data suitable for detailed scientific and environmental analyses. The MBES backscatter data (BL₀ format) are uncompensated and therefore limited to qualitative interpretation; they should not be used for quantitative analyses. These raw backscatter values exhibit systematic intensity variations related to beam angle

geometry, visible as striped patterns perpendicular to survey tracks (Figure 5). Such artifacts restrict applications like

automated acoustic classification or statistical sediment property estimation. However, qualitative visual interpretation for general sediment discrimination remains valid. Pilot processing of a subset area (file 21028_N3355Ac1_BBS.tif) demonstrates that angular compensation substantially reduces these artifacts, indicating that future reprocessing would greatly enhance quantitative utility.

The SSS data are suitable for qualitative seabed characterization, though residual geometric discrepancies (up to ~20 m

locally) require cross-validation with bathymetry for precise geometric applications. Collectively, these datasets provide a robust foundation for scientific research and marine stewardship in Polish marine areas.

### 4.3.2 SSS data quality

The quality of the SSS dataset was assured through a combination of careful survey planning, optimum equipment selection, and rigorous data processing protocols. The surveys were conducted using EdgeTech 4205 SSS systems, operated primarily

at a high frequency of 600 kHz to achieve maximum spatial resolution and image clarity. The SSS devices were towed in a stabilized manner, with the sonar height above the seabed adjusted to the operational range, ensuring consistent swath coverage of up to 65 meters on each side.

Positioning accuracy was maintained using a USBL acoustic positioning system, and in shallow areas, the layback method was applied. These measures minimized navigational errors and ensured that the geolocation of sonar imagery was reliable.

The survey design provided at least 5% overlap between adjacent sonar swaths, and the vessel speed was limited to a maximum of 4.1 knots during SSS operations, guaranteeing high data density with more than 20 measurement points per meter across-track.

Data completeness and resolution were key quality metrics for the SSS dataset. The project specifications required that at least one valid sonar measurement should be present in 95% of the 20×20 cm raster grid cells. This requirement was

consistently satisfied, resulting in a seamless, high-resolution sonar mosaic suitable for detailed seabed characterization and object detection.

The SSS data underwent comprehensive post-processing in SonarWiz software, including correction for navigation errors, gain normalization, de-striping, geometric correlation, and TVG regulation. These steps enhanced image clarity, contrast, and geometric correctness – three critical metrics for SSS data quality. While these processing steps significantly enhanced

image clarity and contrast, it is important to note that geometric corrections cannot completely eliminate all positioning discrepancies inherent to side-scan sonar data acquisition and processing. Residual geometric errors may result from navigation solution variations, acoustic positioning limitations, or complex water column conditions. Despite these inherent limitations, the comprehensive post-processing resulted in high-quality mosaics suitable for qualitative seabed characterization and geomorphological analysis.

High geometric resolution was achieved, enabling the identification and classification of small-scale seabed features, such as boulders, sediment types, and anthropogenic objects. The final mosaics exhibited generally good geometric correctness in

most areas, with high contrast that facilitates the discrimination of objects based on shape, size, and shadow. However, residual geometric discrepancies may persist in localized areas, with documented feature misalignments ranging up to approximately 20 meters in places. These limitations are characteristic of side-scan sonar processing and do not substantially impair qualitative geomorphological and sediment-type interpretation but should be considered when performing precise geometric measurements or object localization. Preliminary geomorphological analysis confirmed that the SSS dataset provided clear differentiation between various seabed types, including cohesive sediment outcrops, scattered boulders, and sandy substrates.

## 5 Data availability

The dataset (Janowski et al., 2025) is available under the following link:

https://doi.org/10.26408/southern-baltic-hydroacoustic-datasets

## 6 Conclusions

The comprehensive mapping of the seafloor in the Polish part of the Southern Baltic Sea has yielded invaluable insights into the region's underwater topography and its properties. This extensive effort, involving high-resolution hydroacoustic measurements, has laid a solid foundation for future research and sustainable management of marine resources. The data acquired through this study will support various scientific, environmental, and economic activities, contributing to the broader goals of the United Nations Decade of Ocean Science for Sustainable Development and the Marine Strategy Framework Directive.

The successful completion of the high-resolution hydroacoustic survey in the southern Baltic Sea represents a significant milestone in marine science and environmental protection. The insights gained from this study will not only advance our understanding of the ocean but also support the development of science-informed policies to ensure a resilient and productive marine ecosystem for future generations. Continued efforts in seafloor mapping and interdisciplinary collaboration will be crucial in addressing the ongoing challenges our oceans are facing and achieving sustainable ocean management.

## Author contributions

Łukasz Janowski: conceptualization, formal analysis, visualization, writing – original draft preparation, writing – review & editing, supervision. Anna Barańska, Aleksandra Bojke, Anna Borecka, Agnieszka Cichowska, Grażyna Dembska, Diana Dziaduch, Sara Foit, Katarzyna Galer-Tatarowicz, Marcin Kalarus, Maria Kubacka, Piotr Pieckiel, Monika Michałek, Ewelina Misiewicz: resources, investigation. Roksana Bona, Patryk Dombrowski, Agnieszka Flasińska, Maciej Kałas, Tomasz Kusio, Emilia Leszczyńska, Aliaksandr Lisimenka, Krzysztof Załęski: resources, investigation, supervision.

Aleksandra Gadzińska, Łukasz Gajewski, Karol Ginał, Urszula Grzywińska, Izabela Górecka, Olha Hruzdieva, Sandra Korczak, Mateusz Kołakowski, Karolina Rogowska, Marcin Sontowski, Marta Szafrańska, Kazimierz Szefler, Anna Tarała, Paweł Wysocki: data curation, software, validation. Juliusz Gajewski, Radosław Opioła: conceptualization, funding acquisition, project administration, methodology, supervision, validation. Edyta Jurkiewicz-Gruszecka, Magdalena Kamińska, Małgorzata Marciniewicz-Mykieta: conceptualization, funding acquisition, project administration, supervision. Natalia Kaczmarek: resources, supervision, validation, project administration. Jarosław Nowak, Radosław Wróblewski: methodology, supervision, validation.

## Competing interests

The contact author has declared that none of the authors has any competing interests.

## Acknowledgements

We extend our gratitude to the dedicated team of approximately 250 individuals who contributed to this project. In particular, we would like to express our sincere thanks to the captains, crews, and scientific staff of the seven vessels for their skilful and efficient cooperation during the work at sea. Their expertise and commitment were instrumental in the successful completion of this endeavour. The project titled: "Mapping of Benthic Habitats of Polish Marine Areas Using the Sonar Mosaicking Method in 2021–2023" was established by the Polish Chief Inspectorate for Environmental Protection within the framework of the State Environmental Monitoring and it was financed from the Polish National Budget. This publication was financially supported by the statutory activities of the Gdynia Maritime University under the projects IM/2025/PI/01 and IM/2025/PZ/03.

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
