# Peer review of "High-resolution hydroacoustic datasets for over 7000 km2 of Southern Baltic"

_Earth System Science Data, 2025_

## Referee Comment (RC2)

**General comments**

Janowski et al. provide a clear description of what appears to be a comprehensive seabed mapping dataset in the Southern Baltic. This is an exciting dataset, and the authors have achieved a high standard of accessibility by making all data open. I believe this is exactly what seabed mapping researchers should be doing to further the field given the great expense and effort required to collect such data, and as we collectively work towards regional and global mapping goals such as Seabed 2030.

The data itself seems to be generally of high quality and usefulness, but some exceptions should be acknowledged. From what I can tell from the manuscript, and from downloading a few samples, the MBES bathymetric data look to be of high quality and resolution – suitable for geomorphic, habitat or other analyses. The authors report that the MBES backscatter were not fully processed, foregoing angular compensation. This is really unfortunate, and limits the usefulness of such an important data source. I cannot understand why the backscatter data were not fully processed, as it seems the authors had access to the necessary tools to do so. If possible, I would encourage the authors to revisit the MBES backscatter at some point and try to update their dataset with an angular-compensated product. The SSS data look to be mostly high quality, but there are some obvious geometric errors that are not mentioned. In fact, the authors state that the SSS data are high quality and free of geometric errors, which is not quite true. See a specific example below.

**Specific comments**

Abstract. Not clear what the difference is between "detailed bathymetric grids at 50 x 50 cm resolution and sonar mosaics at 20 x 20 cm resolution" in the abstract. I believe "sonar mosaics" refers to side scan backscatter, but perhaps make this clear.

Table 1. I'm not sure what is meant by "Seldom MBES and SSS datasets…"

101-102 and onward. Suggest being consistent with use of abbreviations once they have been introduced throughout the manuscript (e.g., "MBES", "SSS").

Figure 1. The quality of this figure could be improved. The base map contains many labels that cannot be read. All labels with the individual survey sheets are too small to be read at normal page size. Legend entries do not need to be complete sentences and could be reduced for conciseness; for example, "The offshore wind energy area excluded from the hydroacoustic study" could just be "Excluded offshore wind area". This is explained previously in the text.

174-175. I am not sure what is meant by "The processing of bathymetric data included the elimination of acoustic noise and imaging of the seabed regarding mean sea level". Does imaging refer to backscatter? Consider rewording/rephrasing.

188. It is a shame that angular corrections were not applied to the MBES backscatter. This greatly reduces the utility of the data for other users. I'm not entirely clear on why this wasn't performed… both the QPS suite and BeamWorx contain functionality for angular correction as part of the backscatter processing, which takes no additional time. Why use proprietary software that cannot

perform a basic AVG correction? This seems like a major shortcoming.

Figure 5. Suggest adding some indication of the backscatter units here – maybe just in the caption. It looks to be possibly just signed 8-bit integer values? Some readers may be confused if expecting to see a dB representation. The use of signed integers and a divergent colour ramp is also a bit odd... This is a unipolar variable being mapped (i.e., low to high values). The convention would be a sequential colour scheme ranging from light to dark. A divergent palette makes it look like the negative and positives are meaningful (they are not).

Figure 5. We see the impact of foregoing the angular correction here. The usefulness of these data for any sort of quantitative analysis is greatly reduced.

253-255, 288-305. In these lines, I believe the quality of the SSS data may be overstated. It is stated that, "The final mosaics exhibited strong geometric correctness, with minimal distortion or artefacts, and high contrast, facilitating the discrimination of objects based on shape, size, and shadow". There are some very obvious geometric discrepancies though, with misalignment of features by at least 20 m (see Figure 1 below). This is not uncommon for sidescan data, but these limitations should really be acknowledged – these are not perfect datasets.

[Figure]

Figure 1. SSS survey OWF_SSS_20cm_1A.

281-282. I'm not sure I fully agree with this statement. The MBES backscatter were not subjected to full angular correction, and probably are not suitable for quantitative geomorphological/geological/habitat mapping analyses. The processing of these data is essentially incomplete.

---

## Author Response (AR1)

**Reviewer 1**

General comments

Janowski et al. provide a clear description of what appears to be a comprehensive seabed mapping dataset in the Southern Baltic. This is an exciting dataset, and the authors have achieved a high standard of accessibility by making all data open. I believe this is exactly what seabed mapping researchers should be doing to further the field given the great expense and effort required to collect such data, and as we collectively work towards regional and global mapping goals such as Seabed 2030.

**We would like to express our sincere thanks to reviewer for such a positive response and acknowledgement of our effort to make the dataset openly accessible. From the beginning, our aim was to enable seabed mapping data gathered in the Southern Baltic to be a source for the scientific community and to feed into wider initiatives such as Seabed 2030. It is obvious that open data sharing is a critical issue in recent marine research due to the enormous resources required to acquire data. We welcome the fact that our approach meets such collective goals and appreciate the reviewer's support for this direction.**

The data itself seems to be generally of high quality and usefulness, but some exceptions should be acknowledged. From what I can tell from the manuscript, and from downloading a few samples, the MBES bathymetric data look to be of high quality and resolution – suitable for geomorphic, habitat or other analyses. The authors report that the MBES backscatter were not fully processed, foregoing angular compensation. This is really unfortunate, and limits the usefulness of such an important data source. I cannot understand why the backscatter data were not fully processed, as it seems the authors had access to the necessary tools to do so. If possible, I would encourage the authors to revisit the MBES backscatter at some point and try to update their dataset with an angular-compensated product. The SSS data look to be mostly high quality, but there are some obvious geometric errors that are not mentioned. In fact, the authors state that the SSS data are high quality and free of geometric errors, which is not quite true. See a specific example below.

**We would like to thank the reviewer for the in-depth review and valuable comments on data quality. We thank for the positive comments regarding the MBES bathymetric data and take note of the issues raised regarding the backscatter and SSS datasets from MBES.**

**Regarding the MBES backscatter, the reviewer correctly identifies that the MBES backscatter data were processed only to the $BL_0$ level without angular compensation. This limitation is acknowledged in the manuscript (Section 3.2.1). The decision was driven by practical constraints and the project requirements: the**

limited project timeline combined with the massive scale of data collection (over 7,000 km$^2$ surveyed across 95,000 km of survey lines). We agree that angular compensation is an important step for improving data usability. At the time of processing, our primary focus was on ensuring timely release of the dataset to support ongoing research and regional mapping efforts.

We agree that angular compensation would increase the suitability of the backscatter data for quantitative sediment classification and habitat mapping. The reviewer's suggestion to revisit and re-process the dataset for fully processed, angular-compensated backscatter is well taken and would constitute a major enhancement to what is already a considerable dataset. This could be addressed in follow-up work, especially since the raw data have been retained and the processing framework is documented. The revised manuscript was improved by clearly spelling this out as a limitation in the technical validation section, along with recommending angular compensation processing as a priority for future updates to the dataset.

We would like to thank the reviewer for pointing out the geometric errors in the SSS data. The remark of the reviewer is valid about the presence of geometric errors in the SSS data. Indeed, while in Section 4.2.2, it is reported that the SSS mosaics exhibited "strong geometric correctness with minimal distortion or artefacts", the manuscript likely overstated the quality since this assertion underplayed the fact that some geometric error remains despite the wide-ranging post-processing workflow.

These processing steps included navigation error correction, geometric correlation, and other quality enhancements using SonarWiz software. However, the revised manuscript benefited from a more balanced assessment that acknowledges residual geometric errors.

Specific comments

Abstract. Not clear what the difference is between "detailed bathymetric grids at 50 x 50 cm resolution and sonar mosaics at 20 x 20 cm resolution" in the abstract. I believe "sonar mosaics" refers to side scan backscatter, but perhaps make this clear.

Thank you for pointing this out. We agree that the distinction between "detailed bathymetric grids at 50 × 50 cm resolution" and "sonar mosaics at 20 × 20 cm resolution" was not sufficiently explained in the abstract. To clarify, the bathymetric grids refer to multibeam echosounder (MBES) depth measurements processed into a continuous surface, while the sonar mosaics refer to side-scan sonar (SSS) backscatter imagery, which provides information on seabed texture and

**composition. We revised the abstract to explicitly state that the sonar mosaics are derived from SSS backscatter data to avoid ambiguity.**

Table 1. I'm not sure what is meant by "Seldom MBES and SSS datasets…"

**Thank you very much for this comment. According to the manuscript, these "seldom" datasets represent data from supplementary or previous surveys that were integrated into the overall project but differ from the primary 2022-2023 survey data, either by having different resolutions or lacking MBES backscatter measurements. To avoid misunderstanding, we rewrote the caption as follows: "MBES and SSS datasets with non-standard resolutions or lack of MBES backscatter, including data from previous surveys (Słupsk Bank and Koszalin Bay) integrated into the overall dataset."**

101-102 and onward. Suggest being consistent with use of abbreviations once they have been introduced throughout the manuscript (e.g., "MBES", "SSS").

**Thank you for pointing this out. We agree that the consistent use of abbreviations makes for better clarity and readability. We have gone through the manuscript with care to ensure that abbreviations, including "MBES" for multibeam echosounder and "SSS" for side-scan sonar, among others, are used consistently after their first introduction, and that full terms are only used where necessary for clarity.**

Figure 1. The quality of this figure could be improved. The base map contains many labels that cannot be read. All labels with the individual survey sheets are too small to be read at normal page size. Legend entries do not need to be complete sentences and could be reduced for conciseness; for example, "The offshore wind energy area excluded from the hydroacoustic study" could just be "Excluded offshore wind area". This is explained previously in the text.

**We thank the reviewer for constructive comments regarding Figure 1. Issues were addressed by replacing the base map with a clean version, increasing the size of all labels to make them more legible, and making the items listed in the legend more concise. For instance, "The offshore wind energy area excluded from the hydroacoustic study" was shortened to "Excluded offshore wind area," with similar changes to other items in the legend. We believe these changes significantly improve the clarity and usability of the figure.**

174-175. I am not sure what is meant by "The processing of bathymetric data included the elimination of acoustic noise and imaging of the seabed regarding mean sea level". Does imaging refer to backscatter? Consider rewording/rephrasing.

**Thank you very much for this comment. We rephrased this sentence to be more clear and informative: "Bathymetric data processing consisted of two primary steps: (1) the removal of acoustic noise and erroneous measurements, and (2) the standardization of all depth values to the PL-EVRF2007-NH vertical reference system referenced to mean sea level."**

188. It is a shame that angular corrections were not applied to the MBES backscatter. This greatly reduces the utility of the data for other users. I'm not entirely clear on why this wasn't performed... both the QPS suite and BeamWorx contain functionality for angular correction as part of the backscatter processing, which takes no additional time. Why use proprietary software that cannot perform a basic AVG correction? This seems like a major shortcoming.

**All MBES backscatter data were processed to $BL_0$ format (raw backscatter data, without angular compensation correction, Schimel et al., 2018) due to limitations in project time and operational constraints. Although angular compensation using the available software (QPS suite, FMGT) would enhance the usefulness of backscatter for quantitative sediment classification, a number of practical issues limited the use of full implementation. First, the acquisition of MBES backscatter was primarily used as a verification tool for sonar imagery rather than as a principal product of the survey. Second, software limitations prevented the concurrent processing of co-located bathymetric and backscatter data stored in proprietary Qinsy *.db formats, requiring sequential processing with different software packages. Third, QPS recommendations to process data in smaller areas to maintain performance resulted in only a limited spatial extent for advanced processing. A pilot application of angular correction to a subset of the data (file 21028_N3355Ac1_BBS.tif) successfully demonstrated significant reductions in both striped and other artifacts and improved image quality. Future work should be directed towards the complete angular compensation processing of the entire backscatter data set as a dedicated follow-up project, significantly increasing the value of this dataset for quantitative studies of sediment and habitat classification applications.**

Figure 5. Suggest adding some indication of the backscatter units here – maybe just in the caption. It looks to be possibly just signed 8-bit integer values? Some readers may be confused if expecting to see a dB representation. The use of signed integers and a divergent colour ramp is also a bit odd... This is a unipolar variable being mapped (i.e.,

low to high values). The convention would be a sequential colour scheme ranging from light to dark. A divergent palette makes it look like the negative and positives are meaningful (they are not).

**Thank you for pointing this out. The backscatter values in Figure 5 are 32-bit floating-point rasters, representing the relative backscatter intensity. These data were created using an in-house backscatter processing package developed for this study. Figure caption updates reflect this change to avoid confusion and provide clarity that these data are not decibel units. We utilized a grayscale sequential color scheme because the data have a unipolar nature ranging from low to high intensity.**

Figure 5. We see the impact of foregoing the angular correction here. The usefulness of these data for any sort of quantitative analysis is greatly reduced.

**Thank you very much for this comment. To answer it, we added a paragraph to the section 4.2.1 (Multibeam data quality) acknowledging this limitation:**

**"Preliminary geomorphological analysis of the MBES data confirmed the ability to distinguish various seafloor features, such as cohesive sediment outcrops, boulders, and sandy substrates. High spatial resolution, precise positioning, and rigorous survey protocols yielded MBES bathymetric data suitable for detailed scientific and environmental analyses. The MBES backscatter data ($BL_0$ format) are uncompensated and therefore limited to qualitative interpretation; they should not be used for quantitative analyses. These raw backscatter values exhibit systematic intensity variations related to beam angle geometry, visible as striped patterns perpendicular to survey tracks (Figure 5). Such artifacts restrict applications like automated acoustic classification or statistical sediment property estimation. However, qualitative visual interpretation for general sediment discrimination remains valid. Pilot processing of a subset area (file 21028_N3355Ac1_BBS.tif) demonstrates that angular compensation substantially reduces these artifacts, indicating that future reprocessing would greatly enhance quantitative utility."**

253-255, 288-305. In these lines, I believe the quality of the SSS data may be overstated. It is stated that, "The final mosaics exhibited strong geometric correctness, with minimal distortion or artefacts, and high contrast, facilitating the discrimination of objects based on shape, size, and shadow". There are some very obvious geometric discrepancies though, with misalignment of features by at least 20 m (see Figure 1 attached). This is not uncommon for sidescan data, but these limitations should really be acknowledged – these are not perfect datasets.

[Figure]

Figure 1. SSS survey OWF_SSS_20cm_1A.

Thank you very much for identifying these critical issues. We have revised the specified lines to shift from overstated quality claims to a more realistic and transparent scientific communication style.

The updated paragraphs in Section 4.2 (corresponding to lines 253-255) now read as follows: "Visual inspections of bathymetric and backscatter maps, as well as SSS mosaics, confirmed the ability to resolve fine-scale seafloor features and sediment types across most of the survey area. The MBES bathymetric data are of high quality and well-suited for quantitative applications including geomorphological analysis, habitat mapping, and detailed seafloor characterization. However, users should note key limitations in the datasets. MBES backscatter data were processed to $BL_0$ format without angular compensation, making them suitable for qualitative sonar verification rather than quantitative sediment analysis. SSS mosaics show good geometric accuracy but include typical side-scan misalignments of up to ~20 meters; they are best used for qualitative geomorphological and habitat mapping, with MBES bathymetry recommended for precise geometry. Taking these limitations into consideration, the datasets are validated for scientific, environmental, and operational applications including qualitative habitat assessment, regional geomorphological analysis, and marine management planning, while supporting further research and improved understanding of the Polish marine areas."

**Moreover, the updated paragraphs in Section 4.1.2 (corresponding to lines 288–305) now read as follows: "The SSS data underwent comprehensive post-processing in SonarWiz software, including correction for navigation errors, gain normalization, de-striping, geometric correlation, and TVG regulation. These steps enhanced image clarity, contrast, and geometric correctness – three critical metrics for SSS data quality. While these processing steps significantly enhanced image clarity and contrast, it is important to note that geometric corrections cannot completely eliminate all positioning discrepancies inherent to side-scan sonar data acquisition and processing. Residual geometric errors may result from navigation solution variations, acoustic positioning limitations, or complex water column conditions. Despite these inherent limitations, the comprehensive post-processing resulted in high-quality mosaics suitable for qualitative seabed characterization and geomorphological analysis.**

**High geometric resolution was achieved, enabling the identification and classification of small-scale seabed features, such as boulders, sediment types, and anthropogenic objects. The final mosaics exhibited generally good geometric correctness in most areas, with high contrast that facilitates the discrimination of objects based on shape, size, and shadow. However, residual geometric discrepancies may persist in localized areas, with documented feature misalignments ranging up to approximately 20 meters in places. These limitations are characteristic of side-scan sonar processing and do not substantially impair qualitative geomorphological and sediment-type interpretation but should be considered when performing precise geometric measurements or object localization. Preliminary geomorphological analysis confirmed that the SSS dataset provided clear differentiation between various seabed types, including cohesive sediment outcrops, scattered boulders, and sandy substrates."**

281-282. I'm not sure I fully agree with this statement. The MBES backscatter were not subjected to full angular correction, and probably are not suitable for quantitative geomorphological/geological/habitat mapping analyses. The processing of these data is essentially incomplete.

**Thank you very much for this comment. We corrected the meaning of the whole paragraph to be more informative: "Preliminary geomorphological analysis of the MBES data confirmed the ability to distinguish various seafloor features, such as cohesive sediment outcrops, boulders, and sandy substrates. High spatial resolution, precise positioning, and rigorous survey protocols yielded MBES bathymetric data suitable for detailed scientific and environmental analyses. The MBES backscatter data ($BL_0$ format) are uncompensated and therefore limited to qualitative interpretation; they should not be used for quantitative analyses. These**

raw backscatter values exhibit systematic intensity variations related to beam angle geometry, visible as striped patterns perpendicular to survey tracks (Figure 5). Such artifacts restrict applications like automated acoustic classification or statistical sediment property estimation. However, qualitative visual interpretation for general sediment discrimination remains valid. Pilot processing of a subset area (file 21028_N3355Ac1_BBS.tif) demonstrates that angular compensation substantially reduces these artifacts, indicating that future reprocessing would greatly enhance quantitative utility."

**Reviewer 2**

1. General Comments

This manuscript presents a highly valuable hydroacoustic dataset covering over 7,300 km$^2$ of the Southern Baltic. It is one of the largest high-resolution mapping initiatives in the region, employing multibeam echosounder (MBES) and side-scan sonar (SSS) surveys under IHO Order 1a standards. The dataset is openly available via DOI and has broad interdisciplinary value for habitat mapping, offshore wind energy, marine archaeology, and environmental monitoring. The dataset is robust, and the article is appropriate for Earth System Science Data. Minor clarifications would strengthen the paper.

**We thank the reviewer for the positive evaluation of the value and significance of the dataset. We are proud that the dataset is one of the largest high-resolution surveys in the region and are pleased to be reassured that it meets technical standards and has a broad range of interdisciplinary uses. We were happy to carefully take into consideration the specific reviewer's technical comments about backscatter processing and SSS geometric accuracy and feel that responding to these points strengthened the manuscript by helping to provide clearer guidance to future users on appropriate applications for each data product. The revisions make certain that, while we are confident in the overall quality and utility of the dataset, we also identify known limitations transparently that will help users make informed decisions regarding their specific application.**

2. Specific Comments

- Novelty and usefulness: The dataset is new, cost-intensive, and unique at this scale. Usefulness across disciplines is clear.

- Appropriateness for ESSD: Focuses on dataset acquisition, validation, and accessibility. Some methods sections are overly verbose.

- Data quality: Meets IHO standards with ≥95% completeness. MBES BS processed only to BL0 seriously hinders the use of this data. Legacy datasets heterogeneous.

- Significance: Excellent uniqueness and usefulness; minor issues with completeness.

- Consistency: No inconsistencies. Figures/tables are of good quality.

- Presentation quality: Well structured but verbose in Methods.

- Reusability: Data in GeoTIFF with metadata ensures long-term reuse.

- Reference: checked.

**We would like to thank the reviewer for the positive evaluation of the novelty, uniqueness, and interdisciplinary utility of the dataset. We appreciate the reviewer's specific recommendations for improvement along three dimensions: 1) condensing the Methods for conciseness while maintaining clarity; 2) clarity on communicating that MBES backscatter data post-processed to $BL_0$ format are not suitable for quantitative analysis; and 3) more explicit discussion of heterogeneity among legacy datasets integrated within the primary 2022-2023 survey. Each of these points is addressed in our revised manuscript.**

3. Specific questions - remarks

- L76 azoic zones : Do such areas exist? How can we know if there is no life?

**Thank you very much for this question. The term "azoic" traditionally means completely devoid of life, but in practice, truly azoic zones are extremely rare in marine environments. We corrected the sentence into the following: "However, certain areas were excluded from the survey: the immediate vicinity of the shore up to the 5-meter isobath, depths greater than 60 meters, military-designated zones, offshore wind energy-designated areas where previous hydroacoustic surveys had been conducted, and areas designated as having minimal or no benthic habitat value according to environmental designations."**

- L119 replace (Iho, 2020) with (IHO, 2020)

**Thank you. We replaced the citation to the following: "(International Hydrographic Organization, 2020)"**

- General question about quality level S-44 1a order : Have you taken measurements in a reference area where the bathymetry and a reference point are known with precision in order to evaluate the error in Z and XY (= classical cross-check test)?

**Thank you very much for this comment. We clarified the manuscript as the following: "To ensure compliance with IHO S-44 Order 1a accuracy standards, the survey vessels underwent regular validation and calibration procedures. Before each measurement mission, all vessels performed system testing and verification using reference points with established coordinates and depths located in Gdańsk Bay near Gdynia port. Specifically, two reference stones with precisely known positions and elevations were used for vertical and horizontal accuracy verification.**

**Additionally, pitch and roll calibration parameters were checked on known reference targets including submerged objects of established position such as the "Desantowiec" landing craft and the Franken shipwreck.**

**Cross-check validation was performed by running control profiles through multiple survey sections to verify line-to-line consistency and identify potential systematic errors. However, comprehensive statistical analysis of all validation data across the entire 7,300 km² survey area was not conducted as part of this dataset publication. The validation procedures performed indicate that the survey met or exceeded IHO S-44 Order 1a requirements, though formal independent verification against external reference surveys was not undertaken."**

- Table 3 Scanning frequency means ping rate?

**Yes. We added the clarification in the table – scanning frequency (ping rate).**

- L185 in proprietary software : Could you clarify that?

**Thank you very much for this comment. Indeed, our team has developed custom software optimized for our processing workflow and data management requirements. This software includes TVG compensation and other standard corrections. We corrected the sentence so it conveys more information in the following way: "Analysis and filtering of MBES backscatter data was performed using custom software developed for this project. Processing included Time-Varying Gain (TVG) compensation (logarithmic attenuation correction based on range, beam number, and intensity), spike cleaning, and outlier removal."**

- L262 and Tables 4 et 5: Shouldn't the Reson 7125 also be considered in Tables 4 and 5?

**Thank you for noting the inconsistency regarding the Reson 7125 in Tables 4 and 5. The Reson 7125 was deployed exclusively on the vessel MIMER for gap-filling and supplementary surveys, representing a small proportion of the overall bathymetric dataset. According to Reviewer's comment, we added the technical specifications of the Reson 7125 to Table 4. For clarity and to avoid redundancy, Table 5 present detailed uncertainty parameters for the two predominant MBES systems (T-50 and R2Sonic 2026) that account for the majority of survey coverage. In order to provide transparency, we added the details in the main text and corrected the table 4 and table 5 caption.**

4. Suggested Additions for Authors

On MBES backscatter limitation: It should be noted that MBES backscatter data were processed only to BL0 format, meaning no angular compensation or full radiometric corrections were applied. While this ensures consistency and timely delivery, it also limits direct quantitative applications (e.g., sediment classification, habitat modelling). Users may need to perform additional post-processing depending on the intended use.

**We thank you for pointing that out and have revised the manuscript to give a more detailed explanation of these limitations and their implications. In particular, we now specify that the backscatter data are in $BL_0$ format, uncompensated, and thus only suitable for qualitative interpretation. We described the beam-angle-dependent artifacts visible in the raw data (Figure 5) and explained why this limits quantitative applications, such as sediment classification or habitat modeling. We added an example of pilot processing on a subset area, which shows that angular compensation significantly improves the data quality, and noted that future reprocessing would enhance the dataset's quantitative utility. Such revisions are important for transparency and to help guide users on what the appropriate uses of this data are and how they can be improved.**

On legacy dataset heterogeneity: Some legacy datasets (e.g., Słupsk Bank, Koszalin Bay) were integrated into the collection. These differ in resolution and do not include MBES backscatter, which introduces some heterogeneity. They are included to ensure coverage but should be used with caution for comparative analyses.

**Thank you for pointing out the heterogeneity that was introduced into this compilation by the inclusion of the legacy datasets from Słupsk Bank and Koszalin Bay. We have revised the text to explicitly mention these resolution differences and to note areas without MBES backscatter. The text now highlights that although such datasets were included for spatial coverage, they can be used only cautiously for comparisons or quantitative analyses. This helps in being transparent and also provides guidance on appropriate interpretation.**